# GAMIFIED CROWD-SOURCING OF HIGH-QUALITY DATA FOR VISUAL FINE-TUNING

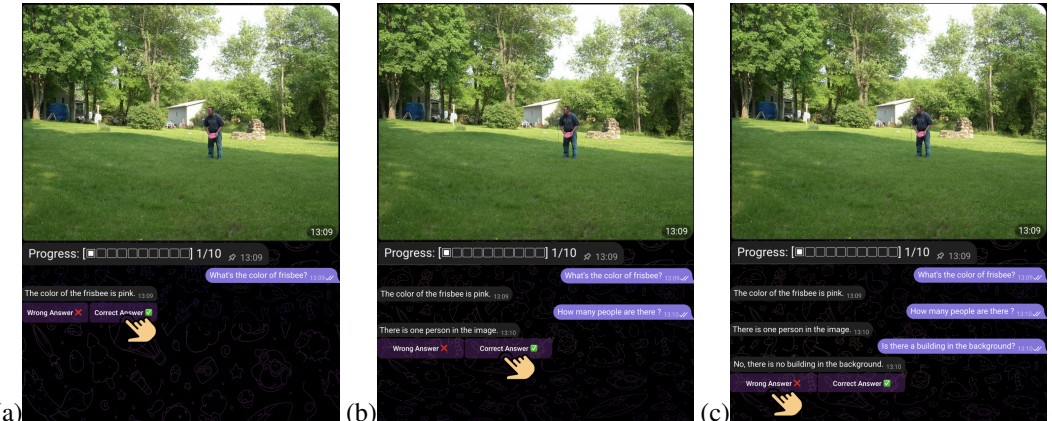

Figure 1: For each image, the goal of the player is to find a question that the AI answers incorrectly: (a) player asks a question and the model answers correctly; (b) player tries again but the model answers right again; (c) finally, the model answers the third question wrong. There are 10 images in a session, 5 tainted and 5 untainted; players don't know which images are tainted. The player moves to the next image after 120 seconds or after choosing "Wrong Answer." At the end of the session, they get 20 points per tainted image where they chose "Wrong Answer" and the model had been instructed to answer incorrectly.

## ABSTRACT

This paper introduces gamified adversarial prompting (GAP), a framework that crowd-sources high-quality data for visual instruction tuning of large multimodal models. GAP transforms the data collection process into an engaging game, incentivizing players to provide fine-grained, challenging questions and answers that target gaps in the model's knowledge. Our contributions include (1) an approach to capture question-answer pairs from humans that directly address weaknesses in a model's knowledge, (2) a method for evaluating and rewarding players that successfully incentivizes them to provide high-quality submissions, and (3) a scalable, gamified platform that succeeds in collecting this data from over 50,000 participants in just a few weeks. Our implementation of GAP has significantly improved the accuracy of a small multimodal model, namely MiniCPM-Llama3-V-2.5-8B, increasing its GPT score from 0.147 to 0.447 on our dataset, approaching the benchmark set by the much larger GPT-4V. Moreover, we demonstrate that the data generated using MiniCPM-Llama3-V-2.5-8B also enhances its performance across other benchmarks, and exhibits cross-model benefits. Specifically, the same data improves the performance of QWEN2-VL-2B and QWEN2-VL-7B on the same multiple benchmarks.

## 1 INTRODUCTION

Visual question answering (VQA) has emerged as a crucial paradigm in AI, extending beyond mere visual interpretation to facilitate broader and deeper understanding in models. Studies demonstrate

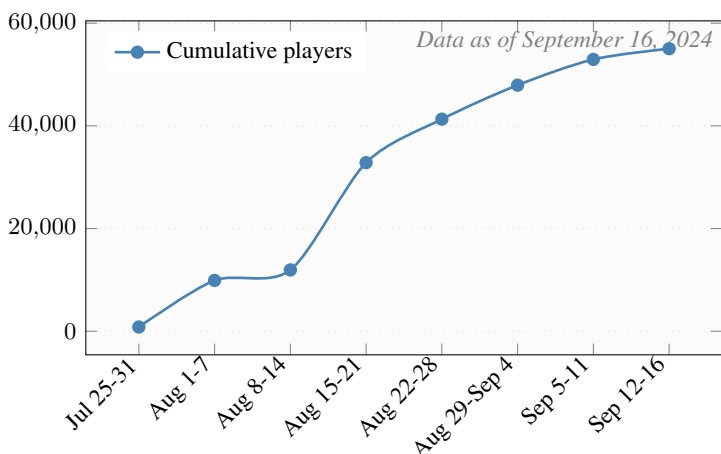

Figure 2: Weekly player growth on our Telegram miniapp.

VQA's potential in enhancing general knowledge acquisition, transfer learning, and complex reasoning skills. Mahdisoltani et al. (2018) showed that pretraining on complex visual-linguistic tasks significantly improves performance across diverse downstream applications, from textual generation to fine-grained classification. The encoding of visual information as language, explored in works like Something-Else (Materzynska et al., 2020; Girdhar & Ramanan, 2019), and more recently by Alayrac et al. (2022), enables models to develop low-level visual skills that support sophisticated reasoning in multimodal contexts.

Large multimodal models (LMMs) have shown impressive capabilities in visual question answering tasks, demonstrating an ability to understand and reason about visual content (Liu et al., 2023a; Dai et al., 2023). However, these models often struggle with fine-grained visual details, factual accuracy, and complex reasoning, particularly when faced with challenging, objective questions (Li et al., 2023c; Yu et al., 2023).

Supervised fine-tuning (SFT) remains a critical yet challenging phase for LLMs and LMMs, especially in visual question answering. SFT includes instruction tuning, where models are optimized to follow natural language instructions. Visual instruction tuning builds on this by fine-tuning models to handle both visual inputs and textual instructions such as questions or commands related to an image. While visual instruction tuning has led to improvements in LMMs, its effectiveness is constrained by the quality and specificity of the training data (Liu et al., 2023a; Li et al., 2023a).

Our gamified adversarial prompting (GAP) method transforms visual instruction tuning by incentivizing players to come up with challenging, fine-grained questions that target specific weaknesses in a given LMM. The GAP approach focuses on improving factual accuracy and reasoning capabilities, particularly for objective, verifiable information more than for subjective preferences or general alignment.

The contributions of this paper include an approach to automatically evaluate and reward player submissions with high accuracy inspired by the reCAPTCHA approach (Ahn et al., 2008). This evaluation scheme ensured we were able to scale GAP to over 50,000 players (Figure 2) in just a few weeks, while ensuring high data quality and player engagement.

## 2 RELATED WORK

**Visual Question Answering (VQA)**   Recent advancements in Visual Question Answering (VQA) have focused on improving model architectures and training methodologies. Our work builds upon these foundations while introducing a novel approach to data collection and model finetuning.

InstructBLIP (Dai et al., 2023) adopted Q-Former to sample from visual tokens for LLM processing. LLaVA (Liu et al., 2023a) introduced visual instruction tuning, using instruction-following data to convert LLMs into multimodal LLMs. Flamingo (Alayrac et al., 2022) and IDEFICS (Laurençon

et al., 2024) used shared decoders for visual-language understanding. Qwen-VL (Bai et al., 2023) employed a three-stage training process to convert LLMs to multimodal models. SPHINX (Gao et al., 2024) adopted multiple visual encoders to enrich visual features. InternLM-Xcomposer (Zhang et al., 2023) achieved state-of-the-art performance using interleaved text-image composition data.

Recently, Yao et al. (2024) introduced MiniCPM-V, a series of efficient LMMs designed for end-side devices. Their work demonstrates the potential for GPT-4V level performance with significantly fewer parameters, while being deployable on mobile phones. This represents a trend towards more efficient and accessible LMMs. We use instruction tuned MiniCPM-Llama3-V-2.5-8B as our base model owing to the small size coupled with high performance.

On other recent advancements in LMMs, Lu et al. (2024) aim for real-world vision-language understanding, while Li et al. (2024a) improve reasoning, OCR, and world knowledge in LMMs. CuMo (Li et al., 2024b) introduced a method for scaling LMMs using mixtures of experts (MoE), incorporating sparse MoE blocks into both the vision encoder and the MLP connector. Recently, Hong et al. (2024) proposed CogVLM2, a family of visual language models that achieve state-of-the-art performance on various image and video understanding benchmarks.

**Adversarial and Challenging Dataset Creation** Creating challenging datasets that expose AI system limitations has been a focus of recent research. SWAG (Zellers et al., 2018), HellaSwag (Zellers et al., 2019), and Social IQa (Sap et al., 2019) introduced large-scale adversarial datasets for commonsense inference. WinoGrande (Sakaguchi et al., 2020) presented an adversarial dataset for commonsense reasoning at scale.

Researchers have also explored algorithmic methods for bias reduction, such as adversarial filtering (Zellers et al., 2018; 2019; Le Bras et al., 2020). These approaches help mitigate bias and reduce dataset artifacts, though they are often applied post-hoc. Our vast and geographically diverse contributor base potentially yields less biased data by incorporating a wide range of global perspectives.

MME (Fu et al., 2023) contains manually annotated instruction-answer pairs to measure perception and cognition abilities on a total of 14 subtasks. DocVQA Mathew et al. (2021), another large dataset, consists of 50,000 questions defined on 12,000+ document images. Liu et al. (2023b) incorporate multiple-choice questions in both English and Chinese versions. In the current work, we focusing only on English. These datasets been created by crowd-sourcing. It is difficult to scale this approach while maintaining quality; our approach is an improvement on this front.

**Benchmarks** The most recent benchmarks include OCRBench (Liu et al., 2024) designed to assess optical character recognition (OCR) capabilities; MMMU (Yue et al., 2024) consisting of multimodal questions from college exams, quizzes, and textbooks spanning 30 subjects and 183 subfields; MM-Vet (Yu et al., 2024) is another benchmark that assesses LMMs on complex tasks requiring the integration of multiple vision-language capabilities; HallusionBench (Guan et al., 2023), challenging benchmark targeted towards LMM hallucinations. All these are available through VLMEvalKit (Duan et al., 2024) We use these benchmarks to evaluate the model finetuned on the dataset generated through GAP.

**Gamification and Human-in-the-Loop Approaches** Our gamified approach to data collection is inspired by works leveraging human interaction to create challenging datasets. CommonsenseQA 2.0 (Talmor et al., 2021) introduced a gamification framework where players compose questions to mislead a rival AI. This approach led to enhanced player engagement and high-quality data collection at scale.

Other relevant human-in-the-loop approaches include Beat-the-AI (Bartolo et al., 2020), which investigated adversarial human annotation for reading comprehension, StrategyQA (Geva et al., 2021), which created a benchmark with implicit reasoning strategies, and Dynabench (Kiela et al., 2021), which proposed dynamic dataset creation. These works demonstrate the value of incorporating human intelligence into the training process.

Fool Me Twice (Eisenschlos et al., 2021) introduced a game for collecting entailment data through multi-player interaction, further illustrating the potential of gamification in dataset creation.

Tools like Shnarch et al. (2022), Tkachenko et al. (2020-2024) and Kim et al. (2024) while not gamified, improve human labelling by using a pretrained AI model.

**Large Scale Data Collection**  Laurençon et al. (2024) has released a massive collection of 50 vision-language datasets covering a wide range of tasks: general visual question answering, counting, captioning, text transcription, document understanding, chart/figure understanding, table understanding, visual reasoning, geometry, spotting differences between 2 images or converting a screenshot to a functional code. Our base model MiniCPM-Llama3-V-2.5-8B (Yao et al., 2024) has been trained on this dataset.

In contrast to Chen et al. (2023) and LAION (2023) who collected detailed image caption data from GPT-4V to augment VQA models, we use a gamified human in the loop approach, maintaining legal and ethical compliance while enhancing transparency in model improvement. This strategy builds our model on a foundation of original, human-verified data, avoiding potential issues associated with AI-generated content. We make part of the data generated during the study publicly available for future research.

**Model Analysis and Robustness**  Recent work has highlighted the importance of thorough model analysis. Studies have shown that while models achieve human parity on many benchmarks, they can be brittle when faced with out-of-domain (Brown et al., 2020; Talmor & Berant, 2019; Fisch et al., 2019) or adversarial examples (Jia & Liang, 2017; Rajpurkar et al., 2018; Yuan et al., 2019; Ribeiro et al., 2018). Yuan et al. (2023); Raina et al. (2024) observed these issues in LLMs as well.

Contrast sets (Gardner et al., 2020) and counterfactual data (Kaushik et al., 2020) have been used to examine model consistency, revealing limitations in current state-of-the-art models. Our dataset generated through GAP can be visualized as a contrast set in relation to the pre-existing knowledge base of an LMM.

## 3  GAMIFIED ADVERSARIAL PROMPTING (GAP)

When it comes to data, quality beats quantity (Zhou et al., 2024; Li et al., 2023b). Given an LMM $M$, an image $i$ and question $q$ related to the image and the expected answer $a$, the model response is denoted $r = M(i, q)$. The tuple $(i, q, a)$ is defined as an **informative sample** for $M$ if the response $r$ is not factually equivalent to the expected answer $a$.

While LMMs are stochastic and generate different responses to the same query, generating a factually incorrect response is usually a sign of a gap in a model's knowledge.

**Gameplay**  One session for one player proceeds as follows:

1. Player sees an image
2. They ask a question which they believe AI would answer incorrectly
3. Player evaluates the AI's response, they have to mark it "Correct" or 'Wrong'
4. If they mark it 'Wrong' they move to the next image, otherwise they can ask more questions. Player automatically moves to the next image after 120 secs.

For every session, a player sees 10 images and has at most 120 seconds for each image. When the player finishes 10 images, or the session resets after 6 hours, they earn points based on their performance. We time-limit every image to create a sense of urgency, while 10 images per session keeps casual gamers motivated, since they can spend limited time on the game but still play for the cash reward.

**Image Datasets**  The GAP framework utilizes two sets of image datasets derived from MS-COCO Lin et al. (2014):

- The **tainted dataset** initially comprises 1000 carefully curated images. The creation process involves selecting images with two or fewer objects, generating detailed descriptions

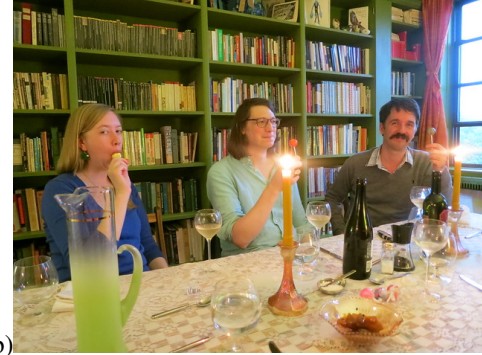

(a)          (b)

Figure 3: (a) Sample image from the tainted dataset, a few easily identifiable items with basic lighting; (b) Sample from the untainted dataset, multiple people and items with complex lighting.

| Dataset (D) | Asked for Incorrect (I) | Model Correct (M) | Human Marked Correct (H) |
|:---:|:---:|:---:|:---:|
| Tainted | ✓ | ✓ | ✓ |
| Tainted | ✓ | ✓ | ✗ |
| Tainted | ✓ | ✗ | ✓ |
| Tainted | ✓ | ✗ | ✗ |
| Tainted | ✗ | ✓ | ✓ |
| Tainted | ✗ | ✓ | ✗ |
| Tainted | ✗ | ✗ | ✓ |
| Tainted | ✗ | ✗ | ✗ |
| | | | |
| Untainted | ✗ | ✓ | ✓ |
| Untainted | ✗ | ✓ | ✗ |
| Untainted | ✗ | ✗ | ✓ |
| Untainted | ✗ | ✗ | ✗ |

Table 1: All possible cases for player interaction: which dataset, whether we asked for an incorrect response, whether the response the player saw was correct, and whether the player marked the response as correct.

using our Base Model (MiniCPM-Llama3-V-2.5-8B), and manually reviewing to select images with complete and error-free descriptions. For these images, the AI model is expected to have complete knowledge.

- The **untainted dataset** includes the remaining, more complex COCO images, used to challenge the LMM.[1]

**Evaluation and Scoring**   In each session a player interacts with five images from the tainted dataset and five from the untainted one. Users are unaware of this split. For each question asked concerning an image from the tainted set, we randomly instruct the model to change its answer subtly, introducing a minor inaccuracy. This process ensures that the incorrect answers are plausible and closely related to the truth, making the challenge more nuanced. Participants are scored based on their ability to identify these deliberate mistakes.

Our goal is to obtain questions on the untainted dataset where the model answers incorrectly. We want to encourage players to ask difficult questions where the model gives an incorrect response, but

---

[1]It could be the case that players learn (maybe unconsciously) the pattern that distinguishes tainted from untainted images. They could then learn that what they do with untainted images does not affect their scores, and then provide bogus input for untainted images. Players will do this if they are motivated by speed, even if they are not hostile. In practice the concern has not materialized as observed with manual evaluations on subsets of data over time. Moreover, after we obtain more than 10 informative data points for an image from the untainted set and the model is fine-tuned on that data, the said image is moved to the tainted set. This helps grow the tainted set and reduces the differences between the two sets.

we also want to disincentivize players from marking correct model responses as incorrect. So our goal is twofold: (1) evaluate which questions from the untainted dataset are incorrectly answered by the model; (2) reward the players appropriately to encourage only desired behavior.

Let $D$, $I$, and $M$ be random variables defined as in Table 1. Let

$$P(M = 1 \mid D = \text{Tainted}, I = 0) = 1 - \varepsilon \tag{1}$$
$$P(M = 0 \mid D = \text{Tainted}, I = 0) = \varepsilon \tag{2}$$
$$P(M = 1 \mid D = \text{Tainted}, I = 1) = \delta \tag{3}$$
$$P(M = 0 \mid D = \text{Tainted}, I = 1) = 1 - \delta \tag{4}$$

where $\varepsilon$ and $\delta$ are small positive numbers such that $\delta < \varepsilon$.

The tainted dataset is constructed to be simpler and more straightforward, as mentioned previously. When not explicitly asked for an incorrect answer ($I = 0$), the model's high accuracy on the tainted dataset is represented by Equation 1. The $\varepsilon$ term accounts for the possibility of errors due to random fluctuations or edge cases. Conversely, the probability of the model answering incorrectly on the tainted dataset when not asked to do so is low (Equation 2).

However, when asked for an incorrect answer ($I = 1$), the model is likely to provide one (Equation 4). There is a small chance the model answers correctly when asked for an incorrect answer (Equation 3). The relationship $\delta < \varepsilon$ indicates that the model is more likely to give an incorrect answer when asked, than to give a correct answer when not asked on the tainted dataset. This reflects the strong influence of the explicit instruction to provide an incorrect answer, overriding the model's natural tendency towards accuracy on the simplified tainted dataset.

Assume that $P(M = 0 \mid H = 0, \text{Untainted}) = P(M = 0 \mid H = 0, \text{Tainted})$. Using Bayes' rule,

$$P(M = 0 \mid H = 0, D = \text{Tainted}) = \frac{P(H = 0 \mid M = 0, D = \text{Tainted}) \cdot P(M = 0 \mid D = \text{Tainted})}{P(H = 0 \mid D = \text{Tainted})} \tag{5}$$

For small $\varepsilon$ and $\delta$,

$$P(M = 0 \mid D = \text{Tainted}) \approx P(I = 1 \mid D = \text{Tainted}). \tag{6}$$

For the denominator, we use the law of total probability, dropping D = Tainted for brevity in the right hand side:

$$P(H = 0 \mid D = \text{Tainted}) = P(H = 0 \mid I = 0) \cdot P(I = 0) + P(H = 0 \mid I = 1) \cdot P(I = 1) \tag{7}$$

Combining these results,

$$P(M = 0 \mid H = 0, D = \text{Untainted}) \approx \frac{P(H = 0 \mid I = 1) \cdot P(I = 1)}{P(H = 0 \mid I = 0) \cdot P(I = 0) + P(H = 0 \mid I = 1) \cdot P(I = 1)} \tag{8}$$

We update this measure at every Tainted sample and choose to select a question as an adversarial question whenever $P(M = 0 \mid H = 0, D = \text{Untainted}) > \theta$. In our experiments $\theta = 0.8$ gives good results.

**Reward system** On a tainted image, the player earns 20 points every time they mark a model response incorrect ($H = 0$) and it is actually incorrect ($M = 0$). No reward is given on untainted images since we cannot evaluate those. The expected reward is

$$R = \sum_{n=1}^{5} 20 \cdot P(M = 0 \mid H = 0, D = \text{Tainted}) \sim \sum_{n=1}^{5} 20 \cdot P(I = 1 \mid H = 0, D = \text{Tainted}) \tag{9}$$

using 6 for approximation.

To maintain high levels of engagement and ensure data quality, GAP incorporates a multi-faceted reward system. Self-Determination Theory (SDT) (Deci & Ryan, 2012) identifies two types of motivation: intrinsic and extrinsic. Intrinsic motivation involves doing something for inherent enjoyment or interest, like playing a video game for fun. Extrinsic motivation is driven by external outcomes, such as studying for good grades. The quality of experience and performance can differ significantly between intrinsically and extrinsically motivated behaviors (Kiela et al., 2021; Cheah et al.,

| Metric | Value |
|---|---|
| Inference speed | 75 input text tokens/second |
| Memory footprint | 19 GB GPU memory |
| Throughput | 0.3 requests/second |
| Average request | 250 input tokens, 640x428 pixel image |
| Cost per 1000 tokens | $0.0037 |
| Hardware | AWS g5.xlarge (1 A10G tensor chip, 24GB memory) |

Table 2: Performance characteristics of MiniCPM-Llama3-V-2.5-8B

2022; Kanellopoulou et al., 2020). Intrinsic motivation leads to better results in the longer term but extrinsic motivation has significant positive impact also.

The game incorporates mechanics to drive player motivation through intrinsic and extrinsic factors:

1. **Experience Points (XP):** Players earn 20 points for each correct answer the AI misses on a tainted dataset, providing immediate feedback.
   - *Extrinsic:* Points act as an external reward.
   - *Intrinsic:* Points foster a sense of competence and mastery.
2. **Leaderboard:** A weekly leaderboard highlights top performers based on points.
   - *Extrinsic:* Public recognition motivates.
   - *Intrinsic:* Competition satisfies the need for relatedness.
3. **Web3 Airdrops:** Future cryptocurrency rewards incentivize high-quality contributions.
   - *Extrinsic:* Clear external incentives.
   - *Intrinsic:* Appeal to players interested in Web3 innovation.
4. **Cash Prizes:** Top-ranking players receive $150 in cash, with rewards for both the top 3 and 3 random players from the top 10.
   - *Extrinsic:* Pure cash incentives.
   - *Intrinsic:* Random prizes create hope and excitement.

This system encourages both short-term engagement and long-term commitment, aiming to gradually transition to token-based rewards with in-game mechanics like levels, NFTs, and guilds. While crypto-native players initially prefer cash and token rewards, leaderboard ranking also drives engagement, even without immediate financial gains.

## 4    EXPERIMENTAL SETUP

We launched our platform as a Telegram miniapp, engaging over 50,000 participants within weeks, without traditional marketing. This rapid growth highlights Telegram's effectiveness for player acquisition in the crypto space. The success of tap-to-earn games like *Catizen* (30 million players) and *Hamster Kombat* (300 million players) further demonstrates the platform's potential. Telegram was chosen over Discord for its larger crypto-native user base and stronger viral growth potential for blockchain projects.

**Baseline Model**    We selected MiniCPM-Llama3-V-2.5-8B as our baseline model due to its state-of-the-art performance in multimodal tasks, cost-efficiency, scalability, and flexible Apache 2.0 licensing. The model demonstrates comparable capabilities to LLaVA (Liu et al., 2023a), Phi-3 (Abdin et al., 2024), and Qwen-VL (Bai et al., 2023; Wang et al., 2024), particularly in single-image understanding and visual question answering. Table 2 summarizes the key characteristics of this model.

The model's efficient resource utilization enables effective horizontal scaling, making it suitable for large-scale experiments and potential real-world applications. These characteristics provide a strong foundation for evaluating improvements in multimodal language understanding and generation tasks while considering practical deployment constraints.

**GAP-VQA Dataset and Experiment Design**   To create GAP-VQA, we randomly sampled $3,683$ question-image pairs, filtering them using a threshold $\theta = 0.8$ (Equation 3) to ensure a high proportion of adversarial examples. While this filtering may include some non-adversarial questions, these do not negatively impact the model; instead, they contribute to the overall diversity of the dataset while having a less pronounced benefit.

Our labeling process combines human expertise with AI refinement to produce high-quality, accurate answers for these visual questions. Initially, trained in-house labeling experts provide answers based on the given images, ensuring a high baseline of accuracy and leveraging human understanding of context and nuance. Subsequently, we employ Llama 3.1 8B (Dubey et al., 2024) to refine these answers, correcting errors, improving clarity, and maintaining consistency across the dataset.

GAP-VQA is divided into two subsets: GAP-VQA-train, consisting of 2,683 question-answer pairs, and GAP-VQA-val, comprising 1,000 question-answer pairs. We carry out three separate types of evaluations:

| Experiment Type | Description |
| --- | --- |
| Same dataset | Fine-tune MiniCPM-Llama3-V-2.5-8B on GAP-VQA-train, then assess accuracy on GAP-VQA-val. This provides a direct measure of how well the model learns to generalize across the GAP-VQA dataset. |
| Cross dataset | Evaluate the fine-tuned model on other benchmarks, such as MME, OCR-Bench, MMBench, and TextVQA. This tests how improved accuracy transfer to standardized multimodal tasks. |
| Cross model | Fine-tune QWEN2-VL-2B and QWEN2-VL-7B on GAP-VQA-train. Evaluate these models on GAP-VQA-val and other benchmarks. This determines whether the GAP dataset for one model benefits another model, indicating its potential for broader AI advancement. |

Table 3: Overview of experiment types

All models are fine-tuned using LoRA (Hu et al., 2022) through Llama Factory (Zheng et al., 2024). We choose LoRA over full fine-tuning of the language and vision encoders to facilitate faster experimentation, and to reduce GPU memory requirements while mitigating catastrophic forgetting.

## 5 RESULTS

The GAP-VQA-train dataset includes a diverse set of questions, divided into 13 groups (Table 8 in the Appendix). These questions assess varying levels of image understanding, from simple tasks like object naming and description to more complex ones like identifying anomalies or explaining cultural events. They evaluate object recognition, spatial reasoning, counting, reading text in images, recognizing emotions, estimating time or season, and interpreting art styles. This broad range of questions helps LMMs improve their generalization capabilities.

**Model Performance on the GAP-VQA-val Dataset**   We evaluated the performance of various models on the GAP-VQA-val dataset using GPT-4 as an evaluator. A comprehensive prompt was used to assess the model's answers based on criteria including existence, position, count, and color accuracy, with scores ranging from 0 to 1, where 1 indicates perfect alignment. The evaluation prompt is detailed in Appendix A.2.

The GAP-VQA dataset was designed to challenge the MiniCPM model by targeting its weaknesses, guiding the fine-tuning process for better alignment. As shown in Table 4, MiniCPM-Llama3-V-2.5-8B achieved major improvement post-fine-tuning, with a score increase of +0.300 (from 0.147 to 0.447). This indicates the dataset's effectiveness in helping capture complex multimodal relationships. In comparison, Qwen2-VL-2B and Qwen2-VL-7B saw more modest improvements of +0.116 and +0.043, respectively, suggesting Qwen2's architecture was less suited for the dataset's complexity. The smaller size of these models, and less aligned pre-training, likely contributed to the weaker improvements.

| Model | Pre-fine-tuning GPT Score | Post-fine-tuning GPT Score | Improvement |
|---|---|---|---|
| GPT-4V (Benchmark) | 0.637 | - | - |
| MiniCPM-Llama3-V-2.5-8B | 0.147 | 0.447 | +0.300 |
| Qwen2-VL-2B | 0.169 | 0.285 | +0.116 |
| Qwen2-VL-7B | 0.207 | 0.250 | +0.043 |

Table 4: Performance on the GAP-VQA-val dataset before and after fine-tuning. GPT-4V is a benchmark and is not fine-tuned.

| Dataset | MiniCPM-Llama3-V-2.5-8B (orig) | MiniCPM-Llama3-V-2.5-8B (finetuned on GAP-VQA train) |
|---|---|---|
| LLaVA Bench | **87.9** | 82.2 |
| OCRBench | 72.4 | **73.1** |
| MME | 2025.61 | **2040.54** |
| RealWorldQA | **0.634** | 0.609 |
| MM-Vet | 51.422 | **51.789** |
| MMBench | **0.752** | 0.7422 |
| HallusionBench | 59.93 | **60.25** |
| TextVQA | 76.63 | **76.966** |
| MMMU val | 0.474 | **0.486** |
| DocVQA | **84.47** | 84.33 |

Table 5: Performance on benchmarks of MiniCPM-Llama3-V-2.5-8B before and after fine-tuning on the AP-VQA train dataset.

**Cross Dataset Evaluation**  Our GAP-VQA approach demonstrates that targeted knowledge gap filling significantly enhances model generalization (Table 5). By training on carefully curated data based on COCO (Lin et al., 2014), the MiniCPM-Llama3-V-2.5-8B model exhibits improved performance across diverse benchmarks, including OCR, hallucination detection, and complex reasoning tasks. This cross-dataset improvement suggests that addressing specific knowledge deficits enables models to develop more robust and transferable skills. The enhanced performance on benchmarks like MME, TextVQA, and MM-Vet, whose contents differ substantially from the training data, underscores the effectiveness of our method in promoting generalization.

**Cross Model Evaluation**  Our cross-model evaluation reveals a compelling phenomenon: addressing knowledge gaps in one model architecture leads to improvements across different model sizes and architectures. This suggests a previously under-explored commonality in multimodal model deficiencies. We observe that GAP-VQA training, initially designed for MiniCPM, significantly enhances both Qwen2-VL-7B and Qwen2-VL-2B models across various benchmarks.

The effectiveness of GAP-VQA across different model architectures suggests fundamental, shared gaps in multimodal understanding. For instance, both 7B and 2B models show marked improvements in LLaVA Bench and MME, with the smaller 2B model exhibiting substantial relative gains. Furthermore, we note that while enhancing performance on targeted tasks, our method preserves or improves capabilities on specialized benchmarks like OCRBench and DocVQA. This indicates that GAP-VQA complements existing model strengths while addressing common weaknesses.

## 6 DISCUSSION

The Gamified Adversarial Prompting (GAP) framework represents a significant advance in the field of AI model improvement. Our results demonstrate substantial improvements in VQA across multiple benchmarks and models. The implications of GAP extend beyond immediate performance gains, suggesting a new paradigm for continuous, scalable AI improvement that leverages human creativity and engagement.

| Dataset | Qwen2-VL-7B (orig) | Qwen2-VL-7B (finetuned on GAP-VQA train) |
|---|---|---|
| LLaVA Bench | 76.7 | **83.6** |
| OCRBench | 86.1 | **86.7** |
| MME | 2318.98 | **2332.71** |
| RealWorldQA | **0.699** | 0.690 |
| MM-Vet | 62.889 | **64.954** |
| MMBench | 0.808 | **0.815** |
| HallusionBench | 68.769 | 68.769 |
| TextVQA | **84.428** | 84.084 |
| MMMU val | 0.524 | **0.527** |
| DocVQA | 93.866 | **94.038** |

Table 6: Performance Qwen2-VL-7B before and after fine-tuning on GAP-VQA train dataset.

| Dataset | Qwen2-VL-2B (orig) | Qwen2-VL-2B (finetuned on GAP-VQA train) |
|---|---|---|
| LLaVA Bench | 52.6 | **57.9** |
| OCRBench | 81.2 | **81.4** |
| MME | 1881.92 | **1962.75** |
| RealWorldQA | **0.626** | 0.6156 |
| MM-Vet | 51.146 | **52.43** |
| MMBench | 0.729 | **0.732** |
| HallusionBench | 61.619 | **62.99** |
| TextVQA | 79.824 | **80.074** |
| MMMU val | 0.414 | **0.448** |
| DocVQA | 89.26 | **89.36** |

Table 7: Performance of Qwen2-VL-2B before and after fine-tuning on GAP-VQA train dataset.

GAP offers advantages over methods that rely on AI self-assessment or using one AI model to evaluate another. Such approaches can be efficient, but they risk perpetuating or even amplifying existing biases and errors. Furthermore, in cases where AI models are trained using outputs from other models without proper attribution or permission, ethical concerns arise regarding the appropriation of intellectual property. Our human-in-the-loop approach sidesteps these issues by leveraging human cognition and diverse perspectives to drive model improvement. By relying on human evaluation within a gamified framework, we respect legal and ethical boundaries while providing a more transparent method for model improvement. This ensures that our model's growth is built on a foundation of original, human-verified data, rather than potentially restricted or problematic AI-generated content.

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

# A APPENDIX

## A.1 PLAYER PARTICIPATION AND PERFORMANCE

Figure 4 reveals key insights into user participation patterns. 69.68% of accounts show no weekly participation, likely including airdrop farming bots—automated programs common in blockchain projects that perform minimal actions to qualify for potential rewards. Despite this, 30.32% of users actively engage at least once per week, with 2.68% participating in multiple weekly sessions.

Figure 4(b) demonstrates that when users do participate, they overwhelmingly interact with all 10 images in a session, as evidenced by the pronounced spike at the 10-image mark. This suggests that the game design effectively encourages thorough engagement once a session begins.

The active user base, while smaller, provides valuable data on the game's appeal and effectiveness. The high completion rate of sessions and the presence of repeat participants indicate strong engagement among active users, showcasing the game's potential to retain and involve players consistently.

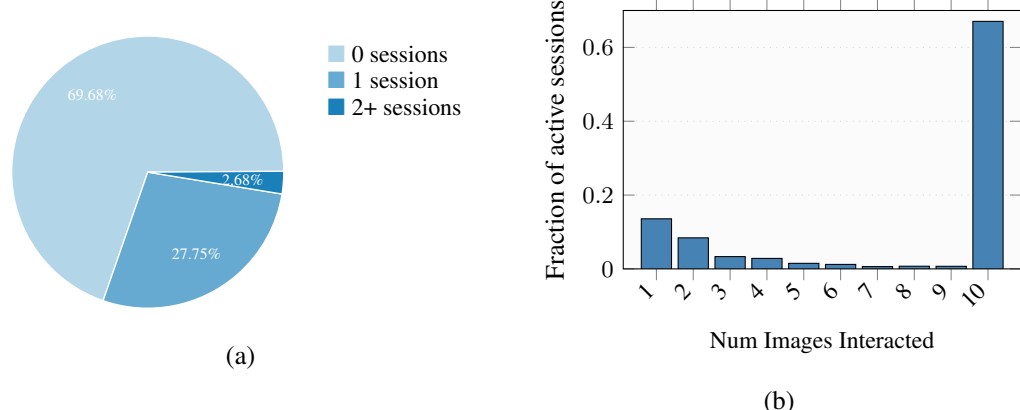

Figure 4: (a) Distribution of user participation in sessions per week. (b) Distribution of the number of images interacted per session.

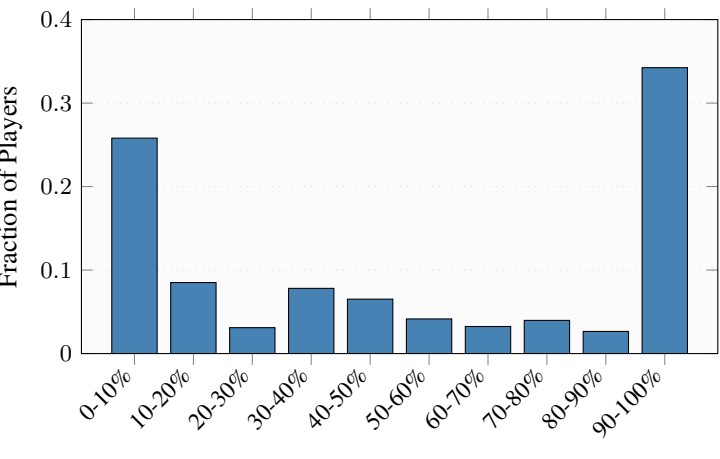

Figure 5: Figure illustrates the bimodal distribution of $P(M = 0|H = 0, D = \text{Tainted})$, revealing that participants tend to cluster at either high or low accuracy levels, as evaluated on the tainted dataset, with fewer achieving mid-range performance

Figure 5 presents a striking bimodal distribution of participant accuracy on the tainted dataset. A substantial 35% of participants achieved exceptional accuracy in the 90-100% range, demonstrating

the task's solvability and the existence of highly effective strategies. Conversely, approximately 26% scored in the 0-10% range, indicating the task's ability to effectively differentiate skill levels. This U-shaped distribution, with peaks at both extremes and lower representation in middle ranges, suggests an optimal difficulty balance.

## A.2 GPT-4V EVALUATION

We used the following prompt to evaluate answers on GAP-VQA-val using GPT-4V:

> Please evaluate the model's answer based on the following criteria compared to the correct answer:
>
>   1. Correct Answer: "{premise}"
>   2. Model Answer: "{hypothesis}"
>
> **Criteria for evaluation:**
>
>   - **Existence:** Does the model's answer correctly identify the existence or non-existence of objects or elements described in the correct answer?
>   - **Position:** Does the model's answer accurately describe the position or location of objects or elements as stated in the correct answer?
>   - **Count:** Does the model's answer correctly state the number of objects or elements mentioned in the correct answer?
>   - **Color:** Does the model's answer accurately describe the color of objects or elements as indicated in the correct answer?
>
> Assign a score from 0 to 1 based on how well the model's answer meets these criteria:
>
>   - A score of "1" means the model's answer fully meets all criteria, accurately reflecting existence, position, count, and color as described in the correct answer.
>   - A score of "0" means the model's answer fails to meet any of the criteria, showing no alignment with the correct answer.
>   - Scores between "0" and "1" should reflect partial correctness, where the model's answer meets some criteria but not all, or has minor inaccuracies.
>
> Carefully consider each criterion before deciding. What is the appropriate score (between 0 and 1) that best represents the factual correctness of the model's answer? Just return the score as a single number.

## A.3 PLAYER INTERACTION MODEL

The probabilistic model for evaluating user-generated questions designed to identify AI mistakes in image analysis incorporates player ability, image difficulty, time pressure, and fatigue to estimate the probability of a player submitting a question that reveals an AI error.

**Formal Definition** We define our VQA model as a function $f : \mathcal{C} \times \mathcal{Q} \times \Theta \times \Omega \to \mathcal{R}$. Let's examine each space in detail:

  - $\mathcal{C}$ (Context Space): This represents all possible input images or visual contexts. *Example:* $\mathcal{C}$ includes all possible digital images, such as photographs of landmarks, diagrams of scientific concepts, or snapshots of everyday scenes.

  - $\mathcal{Q}$ (Question Space): This encompasses all possible questions that can be asked about the visual contexts. *Example:* $\mathcal{Q}$ includes questions like "What color is the car?", "How many people are in the image?", or "What type of architecture is shown in this building?".

  - $\Theta$ (Parameter Space): This represents all possible configurations of the model's parameters. *Example:* For a neural network, $\Theta$ would include all possible weight and bias values for all layers of the network.

| Category | Example Questions |
|---|---|
| Object Recognition | "What is the object on the table?" 
 "Is there a cat in the picture?" |
| Scene Understanding | "Where is this place?" 
 "Is this an indoor or outdoor scene?" |
| Activity Recognition | "What is the person doing?" 
 "Are the people playing a sport?" |
| Object Attributes | "What color is the car?" 
 "Is the cup made of glass?" |
| Spatial Relationships | "Is the apple next to the book?" 
 "How many people are in front of the bus?" |
| Counting | "How many dogs are in the park?" 
 "How many windows does the building have?" |
| Text Recognition | "What does the sign say?" 
 "Can you read the text on the billboard?" |
| Emotion Recognition | "Is the person happy or sad?" 
 "Does the dog look scared?" |
| Weather and Environment | "Is it raining?" 
 "Is the sky clear or cloudy?" |
| Temporal Inference | "Is this image taken during the day or at night?" 
 "Is it summer or winter in this picture?" |
| Cultural or Social Context | "Is this a traditional wedding?" 
 "What festival is being celebrated?" |
| Art and Aesthetics | "What style of painting is this?" 
 "Is this photograph black and white or colored?" |
| Anomaly Detection | "Is there anything strange in this picture?" 
 "Is there an error in this scene?" |

Table 8: Categories of Visual Question Answering (VQA) Tasks

- $\Omega$ (Probability Space): This captures the stochastic nature of the model, representing all possible random outcomes. *Example:* $\Omega$ could represent the randomness in dropout layers, or the variability in outputs due to temperature scaling in the final softmax layer.

- $\mathcal{R}$ (Response Space): This is the space of all possible model outputs or responses. *Example:* $\mathcal{R}$ could include textual answers like "The car is red", numerical answers like "There are 3 people", or more complex responses like "The building exhibits Gothic architecture".

Additionally, we define:

- $\mathcal{A}$ (Answer Space): A subset of $\mathcal{R}$ that contains all correct answers. *Example:* For the question "What color is the sky?", $\mathcal{A}$ might include {"Blue", "Azure", "Cyan"} depending on the specific image.

- $g : \mathcal{C} \times \mathcal{Q} \to 2^{\mathcal{A}}$ (Correct Answer Function): This function maps a context-question pair to a set of correct answers. *Example:* $g$(image of Eiffel Tower, "Where is this landmark located?") = {"Paris", "France", "The capital of France"}

We define **factual equivalence** $\sim$ as a relation on $\mathcal{R}$. For example:

- "A dozen eggs" $\sim$ "12 eggs"

- "The capital of France" $\sim$ "Paris"

- "H2O" $\sim$ "Water"

An **adversarial sample** is a triplet $(c, q, \theta) \in \mathcal{C} \times \mathcal{Q} \times \Theta$ such that:

$$P(\forall r \in g(c,q), f(c,q;\theta,\omega) \nsim r) > \epsilon \tag{10}$$

where $\omega \in \Omega$ and $\epsilon$ is a threshold probability.

*Example:* For a VQA model, an adversarial sample could be (image of the Eiffel Tower, "In which city is this monument located?", $\theta$) where the model consistently fails to answer "Paris" or its factual equivalents.

An **informative datapoint** is derived from an adversarial sample and consists of $(c, q, A)$ where $A = g(c, q)$.

*Example:* Following the previous example, the corresponding informative datapoint would be (image of the Eiffel Tower, "In which city is this monument located?", {"Paris", "The capital of France"}).

Let $L : \Theta \times \mathcal{C} \times \mathcal{Q} \times 2^{\mathcal{A}} \to \mathbb{R}$ be our loss function. The expected change in loss for a datapoint $(c, q, A)$ is:

$$\Delta L(\theta, c, q, A) = E_{\omega \in \Omega}[L(\theta_{new}, c, q, A) - L(\theta, c, q, A)] \tag{11}$$

where $\theta_{new}$ is the updated parameter after training.

We define the global expected loss as:

$$\mathcal{L}_{global}(\theta) = E_{(c,q) \sim P(c,q)}[L(\theta, c, q, g(c, q))] \tag{12}$$

where $P(c, q)$ is the true distribution of context-question pairs in the domain.

An **informative dataset** $D_I$ is a collection of informative datapoints:

$$D_I = \{(c_i, q_i, A_i) | (c_i, q_i) \in \mathcal{C} \times \mathcal{Q}, A_i = g(c_i, q_i), E(c_i, q_i, A_i) > \tau\} \tag{13}$$

where $\tau$ is a threshold for datapoint effectiveness.

*Example:* An informative dataset might include various challenging VQA pairs, such as:

- (image of DNA double helix, "What molecule is this?", {"DNA", "Deoxyribonucleic acid"})
- (image of a quasar, "What astronomical object is this?", {"Quasar", "Quasi-stellar object"})
- (image of the Mona Lisa, "Who painted this masterpiece?", {"Leonardo da Vinci", "da Vinci"})

Let $D_U$ be an uninformative (random) dataset. The global improvement for each dataset is:

$$\Delta\mathcal{L}_{global,I} = \mathcal{L}_{global}(\theta) - \mathcal{L}_{global}(\theta_I) \tag{14}$$
$$\Delta\mathcal{L}_{global,U} = \mathcal{L}_{global}(\theta) - \mathcal{L}_{global}(\theta_U) \tag{15}$$

We argue that $\Delta\mathcal{L}_{global,I} > \Delta\mathcal{L}_{global,U}$ for the following reasons:

1. **Larger Gradient Magnitude:**

$$E[\|\nabla_\theta L(\theta, c, q, A)\| | (c, q, A) \in D_I] > E[\|\nabla_\theta L(\theta, c, q, A)\| | (c, q, A) \in D_U] \tag{16}$$

2. **Higher Information Gain:**

$$E[D_{KL}(P_{true}(r|c,q)||P_\theta(r|c,q)) | (c, q, A) \in D_I] > E[D_{KL}(P_{true}(r|c,q)||P_\theta(r|c,q)) | (c, q, A) \in D_U] \tag{17}$$

3. **Exploitation of Model Weaknesses:**

$$P(\text{Weakness Addressed}|(c, q, A) \in D_I) > P(\text{Weakness Addressed}|(c, q, A) \in D_U) \tag{18}$$

This formalization demonstrates that training on informative datapoints, as generated by the GAP approach, leads to superior improvements in model performance across the entire input space, not just on the specific datapoints used for training.

**Model Formulation** Let $P(S_{ijk})$ be the probability that question $k$ submitted by player $i$ for image $j$ successfully reveals an AI mistake:

$$P(S_{ijk}) = \sigma(\alpha A_i - \beta D_j + \tau(1 - t_{ijk}/T) - \gamma F_{ij}) \tag{19}$$

where:

- $\sigma(x) = \frac{1}{1+e^{-x}}$ is the logistic function, mapping $\mathbb{R} \to (0, 1)$
- $A_i \in [0, 1]$ is the ability score of player $i$
- $D_j \in [0, 1]$ is the difficulty score of image $j$
- $t_{ijk} \in [0, T]$ is the time taken by player $i$ to ask question $k$ for image $j$
- $T > 0$ is the total allowed time per image (e.g., 120 seconds)
- $F_{ij} \in [0, 1]$ is the fatigue factor for player $i$ when reaching image $j$
- $\alpha, \beta, \tau, \gamma > 0$ are scaling parameters

**Parameter Explanation and Domain Reasoning**

- $A_i \in [0, 1]$: Player ability score. Higher values indicate greater skill in identifying AI mistakes. The [0,1] range allows for intuitive interpretation and comparison between players.
- $D_j \in [0, 1]$: Image difficulty score. Higher values represent more challenging images. The [0,1] range facilitates comparison across images and allows for intuitive difficulty scaling.
- $\alpha, \beta > 0$: Scaling parameters for player ability and image difficulty, respectively. These allow the model to adjust the relative importance of these factors.
- $t_{ijk} \in [0, T]$: Time taken for each question. Bounded by 0 and the maximum allowed time T.
- $T > 0$: Total allowed time per image, a positive constant defining the time limit.
- $\tau > 0$: Time pressure scaling parameter. This global parameter adjusts the impact of time pressure across all players and images.
- $F_{ij} \in [0, 1]$: Fatigue factor. We propose modeling fatigue as:

$$F_{ij} = 1 - e^{-\lambda(j-1)/N} \tag{20}$$

  where $\lambda > 0$ is a fatigue rate parameter and $N$ is the total number of images. This formulation ensures $F_{ij}$ starts at 0 and asymptotically approaches 1 as j increases, capturing the cumulative effect of fatigue.
- $\gamma > 0$: Fatigue scaling parameter, adjusting the overall impact of fatigue on performance.

The term $(\alpha A_i - \beta D_j)$ represents the core interaction between player ability and image difficulty. The time pressure term $\tau(1 - t_{ijk}/T)$ increases the success probability for quicker responses, while the fatigue factor $\gamma F_{ij}$ decreases it as the session progresses.

**Model Fitting on Tainted Dataset**   We fit the model using a tainted dataset where the ground truth about AI mistakes is known. The process involves:

1. Data Collection: For each question in the tainted dataset, we record the player identifier, image identifier, question identifier, success outcome (1 if the question revealed an AI mistake, 0 otherwise), and time taken.

2. Parameter Estimation: We use maximum likelihood estimation to fit $A_i$, $D_j$, $\alpha$, $\beta$, $\tau$, $\gamma$, and $\lambda$. The log-likelihood function is:

$$LL = \sum_i \sum_j \sum_k [S_{ijk} \log(P(S_{ijk})) + (1 - S_{ijk}) \log(1 - P(S_{ijk}))] \tag{21}$$

where $S_{ijk}$ is the observed outcome for question $k$ by player $i$ on image $j$.

3. Model Validation: We employ cross-validation, ROC curve analysis, and calibration plots to assess the model's performance and predictive power.

**Application to Untainted Dataset**   For the untainted dataset, where the ground truth is unknown, we apply the fitted model as follows:

1. Probability Estimation: For each submitted question, we estimate $P(S_{ijk})$ using the fitted model parameters.

2. Question Selection: We select questions that meet a predetermined probability threshold $\theta$:

$$Q_j = \{q_{ijk} | P(S_{ijk}) > \theta\} \tag{22}$$

where $Q_j$ is the set of selected questions for image $j$. This threshold helps focus on the most promising questions.

**Model Dynamics and Refinement**   To maintain relevance as players improve and new images are introduced, we implement:

- Dynamic Player Ability: $A_{i,new} = A_{i,old} + \eta(\text{observed\_performance} - \text{expected\_performance})$, where $\eta > 0$ is a learning rate parameter.

- Image Clustering: Grouping similar images to estimate difficulty scores for new, unseen images.

This adaptive framework enables ongoing evaluation of player performance, identification of potential AI mistakes, and continuous improvement of both the evaluation model and the AI system being tested.

