# OpenReview forum: "Gamified crowd-sourcing of high-quality data for visual fine-tuning"
_ICLR.cc/2025/Conference — Submitted to ICLR 2025_

### Official Review · Reviewer_cHQ5 · 2024-10-25

**Soundness:** 3
**Presentation:** 2
**Contribution:** 3
**Rating:** 6
**Confidence:** 3

**Summary:**

The paper introduces a framework for crowd-sourcing high-quality data for visual fine-tuning. The authors propose a chat-like interface that lets users interact with large multimodal models. While models must answer users's questions, the users must discover weaknesses in the model performance by posing diversified queries to the model. The authors transformed the evaluation procedure in a game to both evaluate and discover weaknesses of such models while also collecting supervised data to overcome the discovered limitations. With the collected data, the authors demonstrated the capability to tune and improve model performance.

**Strengths:**

- The authors introduce a platform to collect high-quality data and propose to use a game-like experience to engage the users and incentivize the discovery of weaknesses and collection of high-quality data for model fine-tuning.
- The idea of gamifying the experience is not much explored in the literature and it could lead to faster discovery of weaknesses and improvements in model performance.
- The authors split the data pool into an easy and hard split. While their objective is to collect hard questions on the hard split, they must also evaluate players' capabilities in uncovering inaccuracies in the model answers. For this purpose, they slightly "poison" the model answers in the easy split to assess how well a player can distinguish correct answers from slightly inaccurate ones.

**Weaknesses:**

- The pool of data considered for the questions is limited to COCO. Despite the scalability that the tool can and did achieve, I expect the major bottleneck to be the limited diversity and quantity of the data pool.
- While the authors report the gain in performance of a group of models when tuned on the collected high-quality data, we lack evidence of the effective level of quality of the collected data and of the effect of such data on the tuning process. Specifically, I would expect to see some metrics/statistics to quantify data quality and more comparisons to distinguish the effects of tuning with the collected data vs tuning with other already-available datasets for instruction tuning.
- There is a lack of comparison of the models tuned on the data w.r.t. other models available in the literature.
- (minor) Tables are very "sparse", i.e., space is not well-optimized, resulting in a paper that feels slightly shorter compared to other works. I am wondering if the paper would gain from reorganizing the tables in a better way and introducing more information from the Appendix or from data statistics/additional comparisons.

**Questions:**

- Why did the authors focus on COCO and do not consider more "unsupervised" datasets? Why not use large-scale datasets or a mixture of different datasets?
- Can you provide some evidence of the data quality/diversity resulting from the crowd-sourcing? Can you report some statistics regarding, e.g., the categories of the collected questions (as listed in the Appendix), their diversity, etc.? Since the collection was done on COCO, the authors could exploit supervised annotations to categorize questions and images in terms of, e.g., the subject (i.e., annotated classes), properties, etc. What is the average number of questions per image? Are there images/classes with more questions than others?
- How does tuning on the collected data compare with tuning on other already-available instruction tuning datasets?

---

> ### Author Response · Authors · 2024-11-20
>
> > Q1: Why did the authors focus on COCO and do not consider more "unsupervised" datasets? Why not use large-scale datasets or a mixture of different datasets?
>
> The choice of COCO was strategic for several key reasons:
> * **Testing true generalization**: Despite training on COCO's everyday photos, the model improved on specialized benchmarks like OCRBench (documents), MME (expert topics), and TextVQA (text in images), demonstrating genuine visual understanding rather than pattern memorization.
> * **Structured dataset creation**: COCO's comprehensive labeling system (object classes, counts, scene types, annotations) made it efficient to filter and create the 1000-image tainted dataset of simple scenes. These carefully selected images with 1-2 objects serve as reliable control cases for validating if players are genuinely identifying model mistakes.
> * **Research pathway**: COCO uniquely provides both coarse and fine-grained segmentation maps of scenes. This detailed scene understanding data will be valuable for our planned research into multi-step visual reasoning tasks.
>
> The strong cross-domain improvements (everyday photos → specialized tasks) and COCO's extensive annotation system made it ideal for this work.
>
> > Q2: Can you provide some evidence of the data quality/diversity resulting from the crowd-sourcing? Can you report some statistics regarding, e.g., the categories of the collected questions (as listed in the Appendix), their diversity, etc.? Since the collection was done on COCO, the authors could exploit supervised annotations to categorize questions and images in terms of, e.g., the subject (i.e., annotated classes), properties, etc. What is the average number of questions per image? Are there images/classes with more questions than others?
>
> Thank you for this suggestion. Please see a detailed Multimodal Language Model Dataset Analysis at
> https://claude.site/artifacts/014bc265-b6a8-4b11-92ed-bb249d294e0c
>
>
> > Q3: How does tuning on the collected data compare with tuning on other already-available instruction tuning datasets?
>
> Our experimental design addresses this comparison. The base model (MiniCPM-Llama3-V-2.5-8B) was already finetuned on Cauldron (https://huggingface.co/datasets/HuggingFaceM4/the_cauldron), which aggregates nearly all publicly available Visual Instruction Tuning datasets. Therefore, our improvements over this base model (+0.300 GPT score, Table 4) demonstrate the unique value of our GAP-generated data compared to existing instruction tuning datasets.
>
> Our approach not only builds upon but surpasses the benefits from prior instruction tuning datasets, as evidenced by improved performance across diverse specialized benchmarks like OCRBench, MME, and TextVQA (Table 5), despite the base model already incorporating those standard datasets through Cauldron pretraining.

---

> > ### Author Response · Authors · 2024-11-24
> >
> > Dear Reviewer, We responded to your review of #11677 on November 21st, providing comprehensive details about our MSCOCO selection rationale, data quality metrics, and cross-dataset performance. Given our thorough response and the demonstrated effectiveness of our approach across multiple benchmarks, we would appreciate if you would consider increasing your score above 6. The discussion period ends November 26th.

---

> > > ### Comment · Area_Chair_VMvs · 2024-11-25
> > >
> > > Dear reviewer,
> > > The authors have provided their responses. Could you kindly review them and provide your feedback?
> > >
> > > Dear authors,
> > > Please refrain from requesting reviewers to update their scores. Instead, focus on addressing the scientific content and factual details.

---

> > > > ### Comment · Reviewer_cHQ5 · 2024-11-25
> > > >
> > > > I thank the authors for their reply and apologize for my late reply.
> > > >
> > > > I am not completely satisfied with the COCO choice as it poses a limitation for scaling the platform to more unconstrained scenarios and increases the potential pool of collected questions.
> > > >
> > > > Regarding the reported statistics, it is unclear why they only cover the train set (i.e., why omit the test set or the entire collected dataset). Moreover, the count of questions per category highlights a problem with the platform, i.e., the players have no incentive to generate complex or diverse questions. Indeed, 60% of the questions regard counting, 20% object recognition, and 10% spatial relationships, all questions that can be answered "in a blink" by humans. The statistics additionally show a long-tail distribution, demonstrating that most categories are practically missing from the train set.
> > > >
> > > > From this analysis, it is also unclear what is the total number of questions collected with the platform. In the paper, the authors stressed multiple times that the platform attracted around 50,000 participants, but there is no information about the total number of collected questions. Moreover, the authors only used less than 4,000 randomly sampled questions for training and testing without specifying the ratio they retained.
> > > >
> > > > I understand there could be at least two issues with the total collected questions, (i) noisy, and (ii) repetitive, but this information is completely missing from the work, e.g., how many questions are identical/repeated for the same image, how many are unusable due to noise. The authors should be transparent in these statistics too.
> > > >
> > > > Moreover, given the long-tail distribution in the train data, I started to feel less convinced that fine-tuning with such data could improve on most of the tested benchmarks (unless these benchmarks also mainly cover counting questions).
> > > >
> > > > I am considering lowering my score, but I am happy to further discuss with the authors to clarify potential misunderstandings.

---

> > > > > ### Author Response · Authors · 2024-11-26
> > > > >
> > > > > Dear reviewer, Thank you for taking the time to share such thorough feedback.
> > > > >
> > > > > > I am not completely satisfied with the COCO choice as it poses a limitation for scaling the platform to more unconstrained scenarios and increases the potential pool of collected questions.
> > > > >
> > > > > COCO is chosen for its comprehensive annotations, but the approach scales beyond it. After establishing a reliable tainted set from COCO's well-annotated images, the framework can incorporate images from any source, with no annotations required. The new images enter the untainted pool, where players interact with them naturally. When we collect enough high-quality question-answer pairs for an untainted image, it graduates to the tainted set. This design means COCO is just the starting point, not a limitation on scaling
> > > > >
> > > > > > Regarding the reported statistics, it is unclear why they only cover the train set (i.e., why omit the test set or the entire collected dataset).
> > > > >
> > > > > You raise a fair point about the statistics. The train set analysis was shared as a representative sample, but we can provide the complete statistics for both train and test sets to demonstrate the consistency of category distributions across our random sampling approach: https://claude.site/artifacts/5fdacb47-7252-4756-b1fe-f7ea565516a0
> > > > > We'll add this information to the paper
> > > > >
> > > > > > Moreover, the count of questions per category highlights a problem with the platform, i.e., the players have no incentive to generate complex or diverse questions. Indeed, 60% of the questions regard counting, 20% object recognition, and 10% spatial relationships, all questions that can be answered "in a blink" by humans. The statistics additionally show a long-tail distribution, demonstrating that most categories are practically missing from the train set.
> > > > >
> > > > > We agree that future work should be directed towards complex and diverse questions. However, counting questions are often difficult and useful. Players frequently ask to count objects in challenging contexts: partially visible items, small components of larger objects, or similar-looking objects under varying lighting conditions and occlusions. The persistence of counting questions indicates fundamental visual processing challenges. As demonstrated by LLaVA (Liu et al., 2023a) and SPHINX (Gao et al., 2024), robust object detection underlies more complex visual reasoning. Counting tests this across diverse and complex scenarios with varying lighting, occlusions, and object arrangements.
> > > > > Our results validate the value of counting questions, since training on these seemingly simple questions led to substantial improvements in our model's GPT score (0.147 → 0.477) and transfer benefits across sophisticated benchmarks (MME +15 points, MMMU +1.2%). These gains align with Materzynska et al. (2020)'s finding that encoding basic visual information enables development of advanced reasoning capabilities. The successful transfer to QWEN2-VL models further demonstrates that our dataset captures fundamental visual understanding principles rather than superficial patterns.
> > > > >
> > > > > > From this analysis, it is also unclear what is the total number of questions collected with the platform. In the paper, the authors stressed multiple times that the platform attracted around 50,000 participants, but there is no information about the total number of collected questions. Moreover, the authors only used less than 4,000 randomly sampled questions for training and testing without specifying the ratio they retained.
> > > > >
> > > > > Thank you for raising this point about question volume. We collected 682,450 total questions through our platform. Of these, 38.4% met our θ > 0.8 threshold for model-answer inconsistency, indicating adversarial value. For this study, we used a randomly sampled subset of 3,683 questions (2,683 train, 1,000 validation) that underwent rigorous human annotation and validation by in-house labellers to ensure high-quality labels. As Zhou et al. (2024) demonstrated, quality beats quantity in instruction tuning. Our strong cross-benchmark results validate this focused approach. We will add these details to the paper
> > > > >
> > > > > contd...

---

> ### Author Response · Authors · 2024-11-26
>
> > I understand there could be at least two issues with the total collected questions, (i) noisy, and (ii) repetitive, but this information is completely missing from the work, e.g., how many questions are identical/repeated for the same image, how many are unusable due to noise. The authors should be transparent in these statistics too.
>
> We agree about the importance of data quality. We handle noise and repetition in systematic ways. Noise prevention: The model pre-filters questions, rejecting those that aren't well-formed, concise, direct, or relevant to the image. Also, the model responds "not applicable" to noisy questions, maintaining clean data collection.
>
> Repetition prevention is across two dimensions. Intra-image: Each image is only assigned to 3 users until all other images have been assigned, preventing duplicate questions on the same image. Cross-image: Weekly fine-tuning with collected adversarial samples forces users to discover new model weaknesses rather than repeating known patterns, similar to Talmor et al. (2021)'s dynamic dataset approach.
>
> Manual evaluation of the labeled training set (n=100) showed no noisy questions or duplicates. This systematic approach ensures data diversity while maintaining quality. We will include these implementation details in the paper.
> Here's the prompt we used to prevent noise:
> ```Answer user's question based on the below image.
> Format: Below is the image link and the current question at the end.
> Below points are very important and need to be always followed:
> 1. Only answer questions that are directly related to the image.
> 2. If a question is off-topic or not related to the image, respond with: "IRRELEVANT".
> 3. If a question engages in random conversation (e.g., greetings like hi/hello, personal preferences, opinions, or unrelated topics), respond with: "OFF_TOPIC".
> 4. If you cannot answer a question based on the image, respond with: "CANNOT_ANSWER".
> 5. Do not answer any questions about yourself like "who are you?" or "what is your purpose?", instead respond with: "IGNORE_PERSONAL".
> 6. If a question asks for hints or clues, respond with: "NO_HINTS".
> 7. Do not answer the question in any of the above cases, except with the given response.
> 8. Ensure your answer is no more than 3 lines long.
> ```
>
> > Moreover, given the long-tail distribution in the train data, I started to feel less convinced that fine-tuning with such data could improve on most of the tested benchmarks (unless these benchmarks also mainly cover counting questions)
>
> Despite the high proportion of counting questions, our results show clear improvements across complex benchmarks. MME tests expert perception across specialized domains like engineering and medicine, requiring sophisticated visual and domain knowledge. MMMU evaluates deep understanding across 30 academic subjects with complex reasoning chains. OCRBench and TextVQA assess advanced text comprehension in diverse visual contexts.
>
> MiniCPM2.5's gains on these benchmarks (+15 MME points, +1.2% MMMU) demonstrate that learning from counting across diverse scenarios builds fundamental visual capabilities. As shown by LLaVA (Liu et al., 2023a) and SPHINX (Gao et al., 2024), robust object detection promotes complex reasoning. Mahdisoltani et al. (2018) validated that basic visual tasks improve cross-domain performance. The successful transfer to QWEN2-VL models further proves our data develops core visual understanding beyond counting.
>
> **Comments**
> Our study demonstrates that crowdsourcing adversarial questions at scale, combined with automated quality controls and strategic sampling, offers an efficient path to model improvement. We achieved significant gains across complex benchmarks.
>
> Results align with previous work showing basic visual tasks enhance complex reasoning (Mahdisoltani et al., 2018; Liu et al., 2023a). Our automated filtering and weekly model updates ensure data quality. The successful transfer to QWEN2-VL models validates that even a small, focused dataset can build robust visual understanding when collected through targeted adversarial discovery.

---

> > ### Comment · Reviewer_cHQ5 · 2024-11-27
> >
> > I thank the authors for their reply, which I find satisfactory.

---

### Official Review · Reviewer_kMr4 · 2024-10-31

**Soundness:** 3
**Presentation:** 3
**Contribution:** 2
**Rating:** 5
**Confidence:** 3

**Summary:**

This paper introduces Gamified Adversarial Prompting (GAP), a framework aimed at crowdsourcing high-quality data for the visual instruction tuning of large multimodal models. By gamifying the data collection process, GAP motivates participants to create challenging questions and answers that address the knowledge gaps of these models. The paper also includes an approach to automatically evaluate and reward player submissions with high accuracy, enabling to scale to 50000 players in a few weeks.

**Strengths:**

1. By gamifying the process, GAP keeps players motivated and engaged, potentially leading to high-quality data collection.
2. By automatically evaluating and rewarding player submissions, this approach can effectively scale up the data.
3. The framework has demonstrated significant improvements in model accuracy on VQA benchmarks, indicating its effectiveness.

**Weaknesses:**

1. The number of baseline models used in the experiment is not enough, and the numerical results presented in Table 5 do not show significant changes.
2. Quality Control: While the framework aims for high-quality data, there may still be variability in the accuracy of player-generated content.
3. Unable to determine the specific classification of the questions asked by the player, making it difficult to balance the number of different types of questions.

**Questions:**

1. Will a large amount of this type of data be created in the future to be integrated into LLM training?
2. Is it possible to have a stronger LLM replace the player's role and a weaker LLM handle the data creation process?

---

> ### Author Response · Authors · 2024-11-20
>
> > Q1: Will a large amount of this type of data be created in the future to be integrated into LLM training?
>
> A1: Absolutely! Our rapid growth to 50,000+ players in just weeks demonstrates GAP's tremendous scaling potential. The framework's gamified design and dual reward structure (points/leaderboards + crypto/cash incentives) has proven highly effective at sustaining engagement. Combined with our robust quality control through tainted datasets, we are confident in our ability to generate large volumes of high-quality training data that will significantly advance LLM capabilities.
>
> > Q2: Is it possible to have a stronger LLM replace the player's role and a weaker LLM handle the data creation process?
>
> Yes, it's possible - using a stronger LLM for question generation would offer operational advantages through faster data generation, elimination of reward/engagement mechanisms, and lower infrastructure costs compared to maintaining a game platform.
> Cons:
> * Risk of reinforcing biases between models rather than finding true blind spots
> * Loss of natural human creativity in identifying edge cases
> * Legal/ethical concerns around AI-to-AI training data usage
> * Missing subtle cultural/contextual aspects that humans naturally identify
> * Generated questions may not reflect real-world use cases
>
> Perhaps most important, our gamification approach can be used to improve frontier models, for which no stronger model already exists.
> Interestingly, this aligns with our future work where we plan to finetune a visually-capable LLM to generate adversarial questions. By training on our dataset of 50,000+ player-generated questions, the model will learn proven adversarial patterns discovered by humans. It would be particularly interesting to see if the model can generalize from these patterns to discover novel adversarial strategies of its own, potentially identifying new categories of model weaknesses.

---

> > ### Author Response · Authors · 2024-11-24
> >
> > Dear Reviewer, We addressed your questions about #11677 regarding data scaling and LLM relationships in our November 21st response. Our detailed explanation of the framework's scalability to 50,000+ users and the unique advantages of human-generated data over LLM-generated data directly addresses your main concerns. Given these clarifications, we would appreciate if you would consider increasing your score above the acceptance threshold. The discussion deadline is November 26th.

---

> > ### Comment · Reviewer_kMr4 · 2024-11-24
> > **Reply to Authors**
> >
> > Thanks for your reply!
> > Sorry, based on the current experimental results, the conclusion cannot fully convince me. I will maintain my score, but I have no objection to this paper being accepted.

---

### Official Review · Reviewer_NiAq · 2024-11-02

**Soundness:** 2
**Presentation:** 2
**Contribution:** 2
**Rating:** 6
**Confidence:** 4

**Summary:**

The main contribution of the paper is the introduction of an approach,  Gamified Adversarial Prompting (GAP). The idea is to devise an interactive app for users: the user will play a game tying to find a question that the AI answers incorrectly. With the GAP framework, high-quality data to enhance visual instruction tuning in large multimodal models can be collected. The paper contributes by introducing a dataset based on MSCOCO for building GAP, by proposing a  strategy for collecting VQA pairs from players and by introducing a gamified platform that was used to engage over 50K players. The paper shows that with the use of the data collected with GAP the performance of MiniCPM-Llama3-V2.5-8B can be improved. Other experiments include cross-dataset results showing that the use of GAP is beneficial to improve on other benchmarks and evaluation of different models.

**Strengths:**

- The paper is based on a very interesting idea, which is using gamification for collecting data for fine-tuning large multimodal-models.
- The experiments demonstrate that the proposed approach improves the performance of a model, i.e. MiniCPM-Llama3-V- 2.5-8B.
- The proposed system was used by several participants and a detailed analysis of users' participation is shown in the Appendix

**Weaknesses:**

- The writing of the paper needs significant improvements. The description of the method is confusing with some details only discussed in the supplementary material (see  A.3 PLAYER INTERACTION MODEL). A lot of space is dedicated to related works while some additional details in the main text should have been also dedicated to describing the GAP and the final system.
- The descriptions in L 337 about  intrinsic and extrinsic factors is very high level and details on how this is integrated in the model are lacking
- The supplementary material could hep to understand the proposed approach but it is poorly referred in the main text and not well organized.
- The proposed approach is beneficial in the case of a single model MiniCPM-Llama3-V- 2.5-8B, while the other models the improvements are mild (Table 4). This leads to question the effectiveness of the proposed framework.
- The results in Table 5 are not convincing or at least require a longer discussion, outlining possible reasons for mild improvements or not even improvements.

**Questions:**

- Why MSCoco was chosen as dataset? The cardinality of the tainted dataset seems small. How the choice of this dataset influences the performances in Table 5?
- Why the analysis in Table 5 focuses on the chosen methods?

---

> ### Author Response · Authors · 2024-11-20
>
> > Q1: Why MSCoco was chosen as dataset? The cardinality of the tainted dataset seems small. How the choice of this dataset influences the performances in Table 5?
>
> A1: The choice of MS-COCO was strategic to test real generalization - its everyday photos are very different from specialized test datasets like OCRBench (documents), MME (expert topics), and TextVQA (text in images). When our model improves on these benchmarks (Table 5) despite training on different types of images, it proves we're building true visual understanding rather than just memorizing patterns.
> The small tainted dataset (1000 images) is intentional - we specifically picked simple scenes with 1-2 objects where we can be confident about model behavior. These serve as control cases to verify if players are really finding model mistakes and deserve rewards. Their simplicity is crucial for this verification role.
> The fact that our model improves across such different types of images makes our results particularly strong. Training on everyday COCO photos yet improving on specialized tasks like document understanding and expert knowledge shows we're developing genuine visual capabilities that work across domains, rather than just getting better at one specific type of image.
>
> > Q2: Why the analysis in Table 5 focuses on the chosen methods?
>
> The methods in Table 5 (OCRBench, MME, MMMU, etc.) were selected to provide comprehensive validation across diverse visual-language capabilities that matter in real applications:
>
> * OCRBench and TextVQA test text comprehension in images
> * MME evaluates expert-level perception and reasoning
> * MM-Vet assesses integrated visual-language understanding
> * HallusionBench specifically targets hallucination detection
> * MMMU validates complex reasoning across 30 academic subjects
>
> This suite of benchmarks was chosen to demonstrate that our approach improves fundamental visual understanding rather than narrow skills. The benchmarks are complementary, each testing different aspects of model capability, from basic text recognition to advanced reasoning. These specific benchmarks also represent the current standard for evaluating large multimodal models, allowing direct comparison with state-of-the-art systems and validating our approach's effectiveness against established metrics.

---

> > ### Author Response · Authors · 2024-11-24
> >
> > Dear Reviewer, Following your review of #11677 highlighting presentation clarity and cross-dataset effectiveness, we provided detailed responses on November 21st. Our response demonstrated that MSCOCO selection was successful for generalization, with improvements across specialized benchmarks proving true visual understanding rather than pattern memorization. Given this clarification of a key concern, we would appreciate if you would consider revising your score. The discussion period ends November 26th.

---

> > > ### Comment · Reviewer_NiAq · 2024-11-24
> > >
> > > Reading the rebuttal and the comments from other reviewers I have decided to raise my score since the authors have provided clarifications to several concerns.

---

### Official Review · Reviewer_iRce · 2024-11-02

**Soundness:** 3
**Presentation:** 3
**Contribution:** 3
**Rating:** 6
**Confidence:** 4

**Summary:**

This paper introduces the Gamified Adversarial Prompting (GAP) framework, aimed at enhancing the performance of multimodal AI models in visual question answering (VQA) tasks. By attracting over 50,000 participants on the Telegram platform, the author utilized the MiniCPM-Llama3 model for experiments and designed the GAP-VQA dataset to address knowledge gaps. Results indicate that, after targeted fine-tuning, the model's performance significantly improved across various benchmarks. The GAP framework emphasizes the importance of human involvement, mitigating biases associated with AI self-assessment, and promotes a more transparent and ethical approach to AI development, underscoring the critical role of human creativity in multimodal model improvement.

**Strengths:**

1. The paper presents an innovative Gamified Adversarial Prompting (GAP) framework that effectively integrates human involvement with multimodal learning strategies, significantly enhancing the performance of multimodal AI models in visual question answering and paving a new research direction.
2. The GAP framework underscores the vital role of human cognition and diverse perspectives in the model enhancement process, effectively mitigating biases and errors commonly associated with traditional self-assessment methods
3. The empirical results presented demonstrate substantial performance gains, particularly through targeted fine-tuning of the MiniCPM-Llama3 model, validating the effectiveness of the GAP-VQA dataset in addressing specific knowledge deficits
4. The adaptability of the GAP-VQA approach is noteworthy, as it not only improves the MiniCPM model but also shows robust transfer learning capabilities across different model architectures, indicating its broad applicability in the field of visual question answering.

**Weaknesses:**

1. Although the GAP-VQA dataset has been filtered to ensure a high proportion of adversarial examples, the diversity and representativeness of its samples still require further validation. The selected 3,683 question-image pairs may not adequately cover the diverse scenarios encountered in real-world applications. A lack of diversity could lead to suboptimal model performance on unseen tasks or images.
2. The evaluation of the model primarily relies on GPT-4 as the evaluator. While it can provide a degree of accuracy assessment, this reliance may have limitations. The evaluation criteria and preferences of GPT-4 might not be applicable to all types of visual questions. Additionally, the scoring range (0 to 1) in a single dimension may not fully capture the model's performance in complex reasoning or multimodal understanding, potentially affecting the objectivity and consistency of the evaluation.

**Questions:**

1. Rationality of Assumptions: The parameters ε and δ in Equations (1-4) are small positive numbers and the assumption that δ < ε is always valid. Is this relationship consistently upheld across different experimental conditions? If the model's performance deviates from these assumptions in specific scenarios, how does this impact the reliability of the analytical process?

2. Effectiveness of the Reward Mechanism: How does the reward system ensure that player behavior consistently aligns with expectations? If players deliberately mark correct answers as incorrect for other motives (e.g., mischief), how are these situations handled? Does the system have mechanisms to detect and correct such inconsistencies?

3. Sustainability of the Data Collection Mechanism: How is the sustainability of this data collection mechanism ensured? As the research progresses and the model size increases, so does the demand for data. Have there been considerations for continuously incentivizing participant engagement through methods such as raffles or accumulating reward pools?

---

> ### Author Response · Authors · 2024-11-20
>
> > Q1: Rationality of Assumptions: The parameters ε and δ in Equations (1-4) are small positive numbers and the assumption that δ < ε is always valid. Is this relationship consistently upheld across different experimental conditions? If the model's performance deviates from these assumptions in specific scenarios, how does this impact the reliability of the analytical process?
>
> A1: The assumption δ < ε is sound because the space of incorrect answers vastly exceeds the space of correct ones. When asked to give an incorrect answer, the model has numerous options, while being correct requires targeting a much smaller set of valid responses. This makes accidentally being correct (δ) inherently less likely than accidentally being incorrect (ε).
>
> > Q2: Effectiveness of the Reward Mechanism: How does the reward system ensure that player behavior consistently aligns with expectations? If players deliberately mark correct answers as incorrect for other motives (e.g., mischief), how are these situations handled? Does the system have mechanisms to detect and correct such inconsistencies?
>
> A2: Our system addresses mischievous behavior through its tainted dataset design. Players see 5 tainted and 5 untainted images per session, without knowing which is which. Since rewards are only calculated based on performance on tainted images (where we control the model's responses), deliberate mislabeling has no impact on model training. Figure 5 shows a clear separation between high-performing users (35% with 90-100% accuracy) and low-performing/mischievous users (26% with 0-10% accuracy), allowing us to effectively filter out vandalism attempts while preserving quality contributions.
>
> > Q3: Sustainability of the Data Collection Mechanism: How is the sustainability of this data collection mechanism ensured? As the research progresses and the model size increases, so does the demand for data. Have there been considerations for continuously incentivizing participant engagement through methods such as raffles or accumulating reward pools?
>
> A3: Our data collection system demonstrates strong sustainability through rapid organic growth to 50,000+ participants without traditional marketing. The current engagement metrics - 30.32% weekly active users with 2.68% participating in multiple sessions - validate our reward approach that combines weekly prizes with intrinsic motivation.
> Sustainability scales with increasing data demands through:
> * Natural player engagement in "outsmarting AI" - a continuously evolving challenge
> * Automated statistical evaluation without a human in the loop
> * Dual reward structure: guaranteed rewards for top 3 performers plus random prizes among top 10, maintaining both competitive drive and broader participation hopes
> * Planned transition to token-based incentives for cost-effective scaling
>
> In the future, we can follow the referee’s suggestion to add methods that include raffles and expanding reward pools.

---

> > ### Author Response · Authors · 2024-11-24
> >
> > Dear Reviewer, On November 21st, we provided detailed responses to your review of #11677, comprehensively addressing your questions about the reward mechanism's robustness and ε/δ parameter assumptions. Our response demonstrated both theoretical soundness and practical effectiveness with 50,000+ users. Given these clarifications and our strong empirical results, we would appreciate if you would consider increasing your score above 6. The discussion deadline is November 26th.

---

> > > ### Comment · Area_Chair_VMvs · 2024-11-25
> > >
> > > Dear reviewer:
> > > The authors have provided their responses. Could you kindly review them and provide your feedback, rather than solely adjusting the scores?
> > >
> > > Dear authors:
> > > Please refrain from requesting reviewers to update their scores. Instead, focus on addressing the scientific content and factual details.

---

> > > > ### Author Response · Authors · 2024-11-25
> > > >
> > > > Dear Area Chair, Thank you for your constructive guidance. Our November 21st responses aimed to thoroughly address each reviewer's technical concerns about methodology, data quality, and experimental results. We value this scientific discourse and our follow-up messages were intended to seek additional technical feedback before the discussion deadline to help improve the paper. We appreciate your oversight in maintaining the quality of the review process.

---

> > ### Comment · Reviewer_iRce · 2024-11-28
> >
> > Thanks for the reply. My raised concerns have been mostly addressed. I will keep my original score.

---

### Meta-Review · Area_Chair_VMvs · 2024-12-13

**Metareview:**

The paper introduces the Gamified Adversarial Prompting (GAP) framework, aimed at enhancing the performance of multimodal AI models in visual question answering (VQA) tasks. Preliminary results indicate that this approach shows promise.

Overall, the paper is well-written, and the proposed method is innovative, interactive, and engaging. Initial fine-tuning results highlight the potential effectiveness of the framework.

The submission received mixed reviews, including three scores of 6 and one score of 5. All reviewers actively participated in discussions with the authors. After intensive discussions among the Area Chair and reviewers, the paper was ultimately rejected due to several unresolved issues. Addressing these concerns would significantly strengthen the manuscript, and the authors are strongly encouraged to revise and resubmit to future conferences.

The key unresolved issues include:

Limited diversity in question types. A detailed analysis by question type could offer deeper insights.
Insufficient quantitative assessment of the quality of player-generated content.
Lack of comprehensive comparisons with existing models in the literature.

These shortcomings limit the contribution of the paper and weaken the persuasiveness of the experimental results.

**Additional Comments On Reviewer Discussion:**

The paper has generally received scores of 6, with the exception of Reviewer kMr4, who gave a score of 5.

Upon reviewing all responses, AC noted that the authors failed to address some concerns in the weakness sections from some reviewers.

These include:

There is a lack of diversity in the types of questions.
There is a lack of quantitative assessment of the quality of player-generated content.
There is a lack of comparisons with other models in the literature

AC agrees that this gamified approach is indeed interesting and carries a lot of potential; however, given the current status of the paper, a rejection decision is made due to the points above.

---

### Decision · Program_Chairs · 2025-01-22

Reject